# Antigen-Specific Antibody Signature Is Associated with COVID-19 Outcome

**DOI:** 10.3390/v15041018

**Published:** 2023-04-20

**Authors:** Bárbara Batista Salgado, Maele Ferreira Jordão, Thiago Barros do Nascimento de Morais, Danielle Severino Sena da Silva, Ivanildo Vieira Pereira Filho, Wlademir Braga Salgado Sobrinho, Nani Oliveira Carvalho, Rafaella Oliveira dos Santos, Julia Forato, Priscilla Paschoal Barbosa, Daniel A. Toledo-Teixeira, Kerollen Runa Pinto, Ingrid Silva Correia, Isabelle Bezerra Cordeiro, Júlio Nino de Souza Neto, Enedina Nogueira de Assunção, Fernando Fonseca Almeida Val, Gisely Cardoso Melo, Vanderson de Souza Sampaio, Wuelton Marcelo Monteiro, Fabiana Granja, William M. de Souza, Spartaco Astolfi Filho, Jose Luiz Proenca-Modena, Jaila Dias Borges Lalwani, Marcus Vinícius Guimarães de Lacerda, Paulo Afonso Nogueira, Pritesh Lalwani

**Affiliations:** 1Instituto Leônidas e Maria Deane (ILMD), Fiocruz Amazônia, Manaus 69000-000, Brazil; 2Laboratory of Emerging Viruses (LEVE), Department of Genetics, Evolution, Microbiology and Immunology, Insititute of Biology, University of Campinas (UNICAMP), Campinas 13000-000, Brazil; 3Instituto de Ciências Biológicas, Universidade Federal do Amazonas (UFAM), Manaus 69000-000, Brazil; 4Fundação de Medicina Tropical, Doutor Heitor Vieira Dourado (FMT-HVD), Manaus 69000-000, Brazil; 5Centro de Estudos da Biodiversidade, Universidade Federal de Roraima (UFRR), Boa Vista 69300-000, Brazil; 6Virology Research Center, Ribeirão Preto Medical School, University of São Paulo, Ribeirão Preto 14000-000, Brazil; 7Faculdade de Ciências Farmacêuticas, Universidade Federal do Amazonas (UFAM), Manaus 69000-000, Brazil

**Keywords:** nucleocapsid, SARS-CoV-2, antibody isotypes, IgG subclass, COVID-19

## Abstract

Numerous studies have focused on inflammation-related markers to understand COVID-19. In this study, we performed a comparative analysis of spike (S) and nucleocapsid (N) protein-specific IgA, total IgG and IgG subclass response in COVID-19 patients and compared this to their disease outcome. We observed that the SARS-CoV-2 infection elicits a robust IgA and IgG response against the N-terminal (N1) and C-terminal (N3) region of the N protein, whereas we failed to detect IgA antibodies and observed a weak IgG response against the disordered linker region (N2) in COVID-19 patients. N and S protein-specific IgG1, IgG2 and IgG3 response was significantly elevated in hospitalized patients with severe disease compared to outpatients with non-severe disease. IgA and total IgG antibody reactivity gradually increased after the first week of symptoms. Magnitude of RBD-ACE2 blocking antibodies identified in a competitive assay and neutralizing antibodies detected by PRNT assay correlated with disease severity. Generally, the IgA and total IgG response between the discharged and deceased COVID-19 patients was similar. However, significant differences in the ratio of IgG subclass antibodies were observed between discharged and deceased patients, especially towards the disordered linker region of the N protein. Overall, SARS-CoV-2 infection is linked to an elevated blood antibody response in severe patients compared to non-severe patients. Monitoring of antigen-specific serological response could be an important tool to accompany disease progression and improve outcomes.

## 1. Introduction

The ongoing COVID-19 pandemic has caused immense mortality and morbidity and has also placed huge social and economic burdens on society [1]. SARS-CoV-2 vaccines have been effective in reducing mortality and clinical symptoms associated with the infection; however, growing concerns remain on the durability of the immune response and its ability to neutralize emerging SARS-CoV-2 variants of concern (VOC) [2]. The emergence of new variants of concern (VOC) and their dissemination remains a perpetual challenge to control the ongoing pandemic worldwide.

The SARS-CoV-2 genome comprises four structural proteins: spike (S), envelope (E), membrane (M), and nucleocapsid (N) [3,4,5]. The fundamental function of the N protein is to package the viral genome RNA into a long helical ribonucleocapsid (RNP) complex and to participate in the assembly of the virion through its interactions with the viral genome and membrane protein M. The N protein is highly conserved among the CoVs and is one of the most abundant structural proteins in virus-infected cells [6,7]. The N-terminal domain (NTD) is responsible for RNA binding, the linker region comprises the Ser/Arg (SR)-rich (SRD) flexible disordered region, and the C-terminal domain (CTD) has an important role in protein dimerization. The spike (S) protein of SARS-CoV-2 is glycosylated, and plays a key role in the receptor recognition and cell membrane fusion process. It is composed of two subunits: S1 and S2. The S1 subunit contains a receptor-binding domain (RDB) that recognizes and binds to the host receptor angiotensin-converting enzyme 2 (ACE2), while the S2 subunit mediates viral cell membrane fusion by forming a six-helical bundle via the two-heptad repeat domain.

It is well documented that N and S proteins are the dominant antigens of coronaviruses that elicit IgA, IgG, and IgM antibodies [8]. There have been conflicting reports on which of the two proteins is most immunogenic; both full-length and truncated proteins have been used for serological diagnosis of SARS-CoV-2 infection [9]. Curiously, children, who usually develop mild symptoms after SARS-CoV-2 infection, tend to generate a higher cytotoxic and humoral response against N protein [10] than adults. In this study, we performed a comprehensive analysis of full-length and truncated N and S protein-specific IgA, total IgG, and IgG-subclass antibody response, together with RBD-ACE2 competitive assay and SARS-CoV-2 live virus neutralization assay in plasma samples of RT-PCR SARS-CoV-2-positive individuals during the first semester of 2020 in Manaus city in Brazil, one of the cities most affected by COVID-19 worldwide. We used healthy controls together with COVID-19 patients with mild illness recruited in an outpatient clinic and severe hospitalized patients and compared their humoral response. Since numerous studies have looked at the role of inflammatory serum markers to understand COVID-19, in this study we focused on the SARS-CoV-2 antigen-specific antibody response in severe and non-severe patients before vaccination to understand the disease outcomes.

## 2. Materials and Methods

### 2.1. Ethics

This study was conducted in accordance with Brazilian law and the principles of the Declaration of Helsinki. Samples analyzed in this study received ethical clearance and informed consent forms were approved by the Comissão Nacional de Ética em Pesquisa (CONEP; 062.00967/2020 and 3.929.646/2020) prior to study implementation [11]. All patients provided oral and written consent.

### 2.2. Patient Recruitment and Sampling

A total of 141 patients (age ≥ 18 years) were enrolled between April and August 2020 in the city of Manaus. Healthy controls (n = 13) did not present any symptoms in the last 14 days and were negative for SARS-CoV-2 at enrolment. Outpatients (n = 80) were recruited at the UPA (Unidade de Pronto Atendimento, Emergency Care Unit) Campos Sales tertiary clinic and had mild disease without need of hospitalization. Inpatients or hospitalized patients (n = 48 patients with n = 131 samples) were recruited at the Hospital e Pronto-Socorro Delphina Rinaldi Abdel Aziz. Severe hospitalized patients presented with severe symptoms such as dyspnea, respiratory discomfort, peripheral oxygen saturation lower than 94% and severe acute respiratory syndrome (SARS). Hospitalized patients were followed up longitudinally for 15 days after enrolment and sequential blood samples were collected. All COVID-19 patients included in this study were RT-qPCR-positive for SARS-CoV-2. Patient demographic, clinical, and laboratory data were collected and stored using the REDCap (Research Electronic Data Capture) software.

Venous blood was collected in EDTA (ethylenediaminetetraacetic acid) vacutainer (BD Vacutainer) tubes. Blood tubes were transported to the laboratory and immediately centrifuged at 2000 rpm. Plasma was then aliquoted and frozen at −70 °C or lower in a biorepository. All study samples were transported, handled, and tested following good clinical (GCP), laboratory practice (GLP) standards and in accordance with the Brazilian Health Regulatory Agency (ANVISA) regulations.

### 2.3. SARS-CoV-2 Proteins

Codon-optimized full-length nucleocapsid (N) protein (residues: 1–419) of SARS-CoV-2 (GenBank accession number QHD43423.2) and its truncated fragments N1 (residues: 1–182), N2 (residues: 115–304) and N3 (residues: 245–419), containing a polyhistidine (6x-His) tag at the N terminus of each protein was expressed in Escherichia coli strain BL21 and purified by affinity and size-exclusion chromatography as previously described [12]. Spike (S) full-length protein was expressed in HEK cells and produced by Prof. Leda Castilho (Universidade Federal do Rio de Janeiro) as previously described [13]. The S1 region of spike protein expressed in HEK cells was purchased from Native Antigen.

### 2.4. Measurement of IgA, IgG and IgG-Subclass Response

Indirect in-house ELISA was performed using SARS-CoV-2 full-length and truncated proteins to detect IgG or IgA antibodies in plasma samples (Appendix A). For coating, 100 ng of antigen per well in carbonate buffer pH 9.6 was incubated overnight. Next, ELISA plates were blocked with 10% skimmed milk in PBS (phosphate-buffered saline) at room temperature (20–25 °C); plates were then washed and incubated with 1:100 dilution of patient plasma for 90 min. ELISA plates were then washed with 0.5% Tween 20 in PBS (PBST) and anti-human HRP (horseradish peroxidase)-conjugated antibody was added (goat anti-human IgG-1:40,000 (SeraCare-KPL) and rabbit anti-human IgA-1:2000 (Sigma-Aldrich, St. Louis, MO, USA)). The reaction was developed using ultrasensitive TMB (3,3′,5,5′-tetramethylbenzidine) (Thermo Fischer, Waltham, MA, USA). The assay was stopped by adding 3 M sulfuric acid and absorbance measured at 450 nm with a microplate reader (Chameleon V plate reader).

For detection of IgG subclass (IgG1, IgG2, IgG3 and IgG4), ELISA plates were coated with 100 ng of nucleocapsid or spike antigens and incubated with 1:100 dilution of patient plasma. After washing, plates were incubated with anti-human IgG subclass antibodies at 1:1000 dilution (mouse monoclonal anti-human IgG1, IgG2, IgG3 and IgG4; Sigma-Aldrich). After incubation, plates were washed and incubated with 1:1000 goat anti-mouse IgG labeled with HRP. Upon incubation, plates were washed and ultrasensitive TMB (3,3’,5,5’-tetramethylbenzidine) added to the ELISA plates, with enzymatic reaction stopped by adding 3 M sulfuric acid and absorbance measured at 450 nm with a microplate reader.

Optical density of samples was divided by optical density of negative controls (pre-pandemic samples) to obtain reactivity index (RI) values. As previously described, we used pre-pandemic samples and performed ROC (receiver-operating characteristic) analysis to obtain cutoff, sensitivity and specificity values [12].

### 2.5. RBD-ACE2 Competitive Assay

NeutraLISA (Euroimmun) a commercial semiquantitative assay was used as per the manufacturer’s instructions to estimate potentially neutralizing antibodies inhibiting the binding of SARS-CoV-2 S1/RBD to ACE2 (angiotensin-converting enzyme 2) receptors of the host cells. Patient plasma was diluted (1:5) in sample buffer or controls and added together with biotinylated ACE2 protein to the microtiter plates coated with S1-RBD protein. If antibodies that inhibit RBD-ACE2 interaction are present in the patient sample, they will compete for binding S1/RBD proteins together with biotinylated-ACE2. Unbound ACE2 was removed in subsequent washing steps. To detect the bound biotinylated ACE2 to S1/RBD, peroxidase labeled with streptavidin was added together with the conjugate. The intensity of the reaction was inversely proportional to the concentration of antibodies blocking RBD-ACE2 interaction in the sample. Assay results were expressed as percentage inhibition (IH) of RBD-ACE2 interaction: %IH = 100% − (optical density of patient sample × 100%/optical density of blank (mean)). Upper threshold of the normal range (cutoff value) was set at 25% IH as per manufacturer’s instructions.

### 2.6. SARS-CoV-2 Plaque Reduction Neutralization Test

A plaque reduction neutralization test (PRNT) was performed in the BSL-3 facility of the Laboratory of Emerging Viruses (UNICAMP) using a B lineage isolate kindly donated by Prof. Edison Durigon (SARS-CoV-2/SP02.2020, GenBank accession number MT126808) isolated from a sample collected on 28 February 2020 in Brazil, as previously described [14]. In brief, 2-fold serial dilutions of plasma samples were incubated with a solution of 300 PFU of SARS-CoV-2 for 1 h at 37 °C. The mixtures were added to Vero cell monolayers in 24-well plates and incubated for 1 h at 37 °C in 5% CO_2_. After viral adsorption, cells were washed with PBS and covered with 1 mL of semisolid medium (DMEM 1×, 1% carboxymethylcellulose, 5% inactivated fetal bovine serum (FBS) and 1% penicillin–streptomycin) and maintained for 3 days at 37 °C in a 5% CO_2_ atmosphere. Then, semisolid media were removed, and cells were fixed with 2 mL of 8% paraformaldehyde solution for 1 h and stained with 1% methylene blue (Sigma-Aldrich) for 30 min. Plaque reduction was calculated for each sample by comparing the number of plaques in wells inoculated with SARS-CoV-2 isolate without plasma samples. Nonlinear regression curve fit of three parameters was performed in GraphPad Prism software (v9.1.2 for Mac OS) to determine mean sera neutralizing antibody titers (50% neutralization testing [PRNT50]). Results were calculated as an average of two independent experiments, each performed using technical duplicates.

### 2.7. Statistics and Data Analysis

Chi-squared tests or Fisher’s exact for two-by-two contingency tables were used to examine the statistical significance and associations between study variables. One-way ANOVA test with Tukey’s post hoc test or two-way ANOVA with Šidák post hoc correction for multiple comparisons was used. Spearman correlation was performed for study variables to understand correlation. All statistical analysis was performed using GraphPad Prism software (v9.1.2 for Mac OS).

## 3. Results

### 3.1. Patient Demographics

A total of 128 individuals with a nasopharyngeal swab positive for SARS-CoV-2 by real-time RT-PCR were included in this study (Appendix A). This included 80 symptomatic patients with or without fever attending an outpatient clinic; samples were collected only at enrolment. Hospitalized patients diagnosed with SARS were longitudinally followed for 15 days after admission. Among the 48 inpatients included in the study, 24 were discharged and 24 succumbed to the disease. Appendix A summarizes the demographic characteristics of the study population, associated comorbidities and clinical symptoms at recruitment. A significant decrease in peripheral leukocytes was observed in inpatients compared to outpatients. Severe hospitalized patients had elevated urea, LDH and C-reactive protein levels (Appendix A). Median days with symptoms before hospitalization were similar for discharged or deceased patients; 20% of discharged patients were admitted to intensive care unit (ICU) and 20% required mechanical ventilation, whereas 95% of the deceased patients required mechanical ventilation (Appendix A).

### 3.2. Robust Humoral Response against the Nucleocapsid and Spike Full-Length, and Truncated Proteins

Full-length and truncated N and S SARS-CoV-2 proteins (Figure 1A) were used in this study to assess humoral response among COVID-19 patients (Figure 1B and Appendix A). Antigenicity of the N, N1, N2 and N3 proteins was confirmed by Western blot assay as described in Appendix A. COVID-19-positive patients had significantly elevated IgA, IgG and IgG-subclass response towards the SARS-CoV-2-specific full-length and truncated proteins (Figure 1C and Appendix A). Robust IgA response was observed towards the C-terminal domain (N3) of N protein compared to full-length N, N-terminal (N1) and linker region (N2). We failed to detect N2-specific IgA response at 1:100 plasma dilution. Lower plasma dilution (1:5 or 1:10) failed to detect anti-N2 IgA response (Appendix A). S-specific IgA response was elevated among COVID-19 patients compared to the N-terminal S1 region. Median N3- and S-specific IgA response was similar among the COVID-19 patients. Overall, anti-N3 IgG and IgG-subclass response was significantly higher compared to N, N1 and N2 proteins, whereas anti-S protein IgG response was most robust among all the antigens tested. In the IgG subclass, breadth of the IgG1 response was the highest, followed by IgG2, IgG3 and IgG4 (Figure 1C and Appendix A). Reactivity towards SARS-CoV-2 antigens increased with the number of days after the onset of the disease. Dominant N3 and S protein response increased gradually within the first days after the infection. Notably, three weeks after onset of symptoms, peak humoral IgA and IgG response was observed. SARS-CoV-2-specific IgA response appears to be short-lived compared to the IgG response and starts to decline towards three weeks after onset of symptoms (Figure 1C).

Hospitalized inpatients had a robust IgA, IgG and IgG-subclass response to nucleocapsid and spike proteins compared to outpatients at recruitment (Figure 1D). N2 protein-specific IgA and IgG response was the lowest among all the antigens tested. Upon comparing inpatients based on the disease outcome, we did not observe differences in the IgA response to N, N1, N2 and N3 proteins. However, an elevated IgA response against S and S1 proteins was observed in discharged patients compared to the deceased patients at recruitment. On the other hand, IgG-specific response to N2 and S1 was elevated in deceased compared to discharged inpatients (Figure 1E).

Overall, severe patients showed a steady increase in antibody reactivity index as disease progressed (Figure 2 and Appendix A). Longitudinal follow-up of hospitalized patients illustrated sustained N, N1 and N3 specific IgA response over three weeks post onset of symptoms. S and S1-specific IgA response showed a decline around four weeks after starting symptoms. Among the IgG subclasses, SARS-CoV-2-specific IgG1 and IgG3 was responsible to elevate the total IgG response, whereas IgG4 response was weak among inpatients (Appendix A).

### 3.3. Presence of SARS-CoV-2 Neutralizing Antibodies Does Not Affect Disease Outcome

A competitive assay was set up to estimate potentially neutralizing antibodies that block RBD-ACE2 interactions among COVID-19 patients. We observed significantly elevated blocking antibodies among inpatients compared to outpatients, which steadily increased after the onset of the symptoms (Figure 3A–D). Median inhibitory antibodies were similar among deceased and discharged inpatients at recruitment. Neutralization assay with live virus confirmed RBD-ACE2 competitive assay results (Appendix A). An increase in SARS-CoV-2-specific neutralizing antibodies was directly proportional to the days after onset of symptoms among the inpatients. Longitudinal follow-up of hospitalized patients demonstrated no relation to disease outcome (Figure 3E–H). Appendix A demonstrate the correlation between humoral response to nucleocapsid and spike proteins compared to the RBD-ACE2 inhibitory assay and PRNT.

Next, we compared the peripheral lymphocyte percentage with the humoral response among COVID-19 patients. Lymphocytes were negatively correlated with peripheral antibody response. This negative correlation between lymphocytes and RBD-ACE2 inhibitory antibodies was also evident among outpatients and inpatients (Appendix A). A principal component analysis using antibody reactivity data against nucleocapsid and spike full-length and truncated protein was used to understand the clustering among the different patient groups. We observed a tight cluster among outpatients with mild or moderate diseases; however, discharged or deceased inpatients did not form clear clusters (Appendix A). Overall, this clustering effect is in line with other observations that there is little difference in terms of antibody response among inpatients and control of inflammation among inpatients leads to recovery from disease.

### 3.4. Relationship between IgG Subclass Signature and COVID-19 Outcome

IgG antibody subclass ratio was used to understand disease progression among outpatients and inpatients. Overall, we observed a similar trend in N and S protein-specific IgG2/IgG1, IgG2/IgG3 and IgG1/IgG3 antibody ratios over the follow-up period among non-severe outpatients and hospitalized inpatients (Appendix A). However, over seven days after the start of symptoms, ratios of IgG2/IgG1, IgG2/IgG3 and IgG1/IgG3 S-protein-specific response were significantly lower among inpatients compared to outpatients, whereas the IgG1/IgG3 ratio for N2-specific response was elevated among inpatients compared to outpatients with mild disease (Figure 4A). Only the N2-linker region of nucleocapsid protein demonstrated a significant reduction in IgG2/IgG1, IgG2/IgG3 and IgG1/IgG3 antibody ratios among deceased patients compared to individuals discharged from hospital after a severe COVID-19 infection (Figure 4B).

## 4. Discussion

The clinical role of antibodies in modulating disease severity during infection, duration and persistence of the immune response and the protective role of these antibodies have been key questions since the start of this pandemic. Lately, most studies have focused on the spike protein and neutralizing antibody response against the VOCs. In this study, we observed SARS-CoV-2 infection elevates IgA, IgG subclass and neutralizing antibody response; however, magnitude of response is superior in patients with severe disease compared to those with non-severe disease. Overall, antibody response in hospitalized patients independent of outcome was similar in COVID-19 patients. Longitudinal evaluation of IgG subclass antibody response in hospitalized patients demonstrated significant differences between discharged and deceased patients, especially towards the disordered linker region of N protein. These results demonstrate that monitoring of antigen-specific viral response can be an important tool to accompany disease progression and improve outcomes.

Our results confirm previous findings that IgA and IgG antibodies specific for SARS-CoV-2 antigens typically become detectable in patients’ blood around two weeks after onset of symptoms. Here we also observed a low level of peripheral humoral response to SARS-CoV-2 antigens in outpatients with mild illness compared to severe hospitalized inpatients, consistent with prior publications [15,16,17] and reports of other coronavirus infections [18,19,20].

SARS-CoV-2 has been shown to directly affect the digestive system and infect intestinal epithelial cells. IgA presence in the peripheral plasma samples was directly proportional to the disease severity. A robust antigen-specific IgA was observed early after onset of infection among hospitalized patients along with a robust IgG response. However, outpatients early in the disease had an IgG response, but a weak IgA response. Nasopharyngeal swabs or saliva from outpatients have been shown to have a lower viral load compared to severe patients, which may correlate with higher amounts of viral antigen in mucosal tissue. This apparent high viral load in mucosal tissues might explain a higher antigen-specific IgA presence in the plasma. Here we observed that the C-terminal N protein and full-length S protein were the most immunogenic. Moreover, we observed antigen-specific plasma IgA response declined rapidly compared to IgG response, as shown previously [21,22,23,24]. However, we know very little about the role and mechanism by which IgA antibodies participate in virus clearance or disease pathology.

Nucleocapsid C-terminal protein and S protein-specific IgG response was detected very early in the patients compared to other antigens. Our data are consistent with results reported from a panel of antibody assays applied to single time-point samples from COVID-19 patients who recovered or died of their disease, which found higher values of spike-targeting responses in the convalescents [17,25]. In our study, we observed that the C-terminal N protein domain was more sensitive in identifying COVID-19 patients in the first week of infection. However, the highest sensitivity of IgG detection was three weeks post-onset of disease. This low level of antibody response to most antigens during the first week of infection can affect detection sensitivity among the mild and moderate cases.

We observed an imbalanced IgG subclass response associated with disease severity. Robust IgG1 subclass response was followed by IgG2, IgG3 and IgG4. In our study, among the total COVID-19 patients, the predominant humoral immune response to nucleocapsid protein was IgG1. IgG2 response was dominated by the N3 protein. However, IgG3 response was not directed against specific N or S full-length or truncated proteins. Lles et al. reported the predominant humoral immune response to nucleocapsid was IgG3, whilst against spike it was IgG1 [26], whereas Yates et al. reported IgG1 and IgG3 predominant against spike S1 and S2 proteins, respectively. In a subsequent study from the same lab, they observed IgG1 response was dominant for full-length nucleocapsid and spike proteins, and IgG3 was most dominant against the full-length S [27]. We did not observe any differences between subclasses and antigen among the outpatients with mild illness, but upon comparing antigen-specific IgG1 reactivity among hospitalized patients, full-length, N-terminal and C-terminal nucleocapsid proteins dominated antibody response. N3-specific IgG2 response was the most dominant among the hospitalized patients. Interestingly, our study demonstrated an increased S-specific IgG1 response among hospitalized patients who were discharged compared to deceased patients. Differences between the subclass humoral response might be due to differences in the geographical regions where the study was carried out and the circulating SARS-CoV-2 variant, associated coinfection among hospitalized patients and composition of the patient study groups.

Neutralization, RBD-ACE2 blocking, and S1-specific IgG were all highly correlated in patients with high antibody levels, but RBD-ACE2 blocking was less sensitive than the neutralization assay, potentially because of antibodies that can neutralize by binding to non-RBD regions of the spike, or lower affinity antibodies that can neutralize in the cell culture assay but do not compete as well with binding of ACE2 under the blocking assay conditions [17]. It is currently unclear, however, which of these assays will be the best predictor of in vivo immunological protection from SARS-CoV-2 infection. In the detailed serological time courses of the hospitalized patients in this study, it was evident that the patterns of antibody responses could not fully explain patient outcomes, including death. Similarly, individuals with moderate antibody production were seen across the full spectrum of inpatient disease severity, and many patients who died of their disease generated high levels of antibodies, RBD-ACE2 blocking activity, and neutralizing titers. Differences between individuals in other aspects of the immune response or disease course, such as production of inflammatory mediators, T cell responses, host cell and tissue vulnerability to the damage during viral infection, underlying comorbidities, and secondary bacterial or fungal infections, are all likely to contribute to patient outcomes.

Our results confirm a reduced peripheral blood lymphocyte count among hospitalized COVID-19 patients compared to patients with mild illness. A negative correlation between lymphocyte and peripheral humoral response for antibody isotypes and RBD-ACE2 blocking antibodies was noted. Despite low lymphocyte counts, severe patients had several-fold higher responses compared to outpatients with higher lymphocyte count. Among the lymphocytes, CD3+ T cells, CD4+ T cells, CD8+ T cells, and natural killer cells have been shown to significantly decrease in patients with COVID-19. In contrast, the decrease in B cell counts among severe COVID-19 patients is not as consistently observed as the decrease in T cell counts [28,29]. Probably, the heightened humoral response in severe patients could be due to the high availability of virus antigen since the start of the disease, as observed by high RT-PCR viral loads among severe patients or robust B cell stimulation before the lymphocyte decline among severe patients. However, the relationship between low lymphocyte counts and a robust B cell memory response induced during primary infection and the contribution of different aspects of immune memory to the protection against SARS-CoV-2 in humans remains unclear.

Inflammation-related soluble blood markers such as IL-6 have been consistently described as a marker for COVID-19 severity. In our study, we observed nucleocapsid linker region-specific IgG3 antibody response was considerably altered between hospitalized patients and lower IgG2/IgG3 and IgG1/IgG3 antibody ratio was correlated with mortality. Previous studies have not investigated antibodies specifically directed towards this region of the nucleocapsid protein. However, although most immune responses directed against protein antigens are of the IgG1 subclass, IgG3 responses can also dominate, especially early in the immune response. Upon immunoglobulin class switching to IgG1 or IgG2 or IgG4, descendant cells cannot make IgG3, as the IgG3-heavy chain locus has then been removed from the genome. Ideally, IgG3+ B-cells can however class switch further to IgG1 or to any other IgG class. We suggest that evolutionarily host survival might be determined by metabolic and immune regulatory choices during difficult times, during excessive inflammation and when immune regulation is uncontrolled hosts responds by producing IgG3 antibody response towards dominant and non-dominant protein epitopes for its survival. By such intervention, the recovery of the patient with dysregulated immune system responses might be sped up and the fitness of an individual efficiently restored upon survival. In COVID-19, we observed hospitalized patients that succumbed to the infection preferentially produced higher levels of antigen-specific IgG3 antibodies compared to IgG1 antibodies. These bioenergetics and dynamics of antigen-specific immune response can be used to monitor disease progression and improve disease outcome.

One of the limitations of our study is that we did not test for IgM responses, since we observed higher cross-reactivity with pre-pandemic dengue positive sample during our assay standardization (data not shown). We detected very early IgA and IgG responses and the classical class switch from IgM to IgG has not been evident among the SARS-CoV-2 infection compared to other arboviral infections endemic in the study region. We could not study the B cell subsets in peripheral blood by flow cytometry to complement our serological analysis. We were not able to perform a longitudinal follow-up of outpatients as we did for inpatients due to logistical challenges at the peak of the pandemic. Follow-up of study participants beyond the study period could bring more information on the persistence of antibody responses for the different immunogenic regions of SARS-CoV-2 antigens. Additionally, we cannot rule out previous infection with other coronaviruses among our study population or medication used during treatment, which may have influenced the antibody responses. Observed decreases in IgA antibody levels do not necessarily indicate a complete loss of immunity, and local memory response in the mucosa might prevent and impede SARS-CoV-2 infection upon re-exposure.

In conclusion, the C-terminal nucleocapsid region of SARS-CoV-2 is immunogenic and might be suitable for seroprevalence and diagnostic purposes. Additionally, evaluation of the IgG subclass response towards disorganized nucleocapsid region can be suitable for monitoring disease outcome; however, further studies are essential in other cohorts to confirm these pilot results. Hence, further detailed study of the generation of memory B cell populations, short- or long-lived plasma cells, and T cell memory to SARS-CoV-2 as well as other coronaviruses would clarify some of these key mechanistic points. Our results complement previous studies and provide a potential use for serological assays for disease monitoring to improve the management of COVID-19 patients by identification of high-risk patients and allocation of appropriate health-care resources.

## Figures and Tables

**Figure 1 viruses-15-01018-f001:**
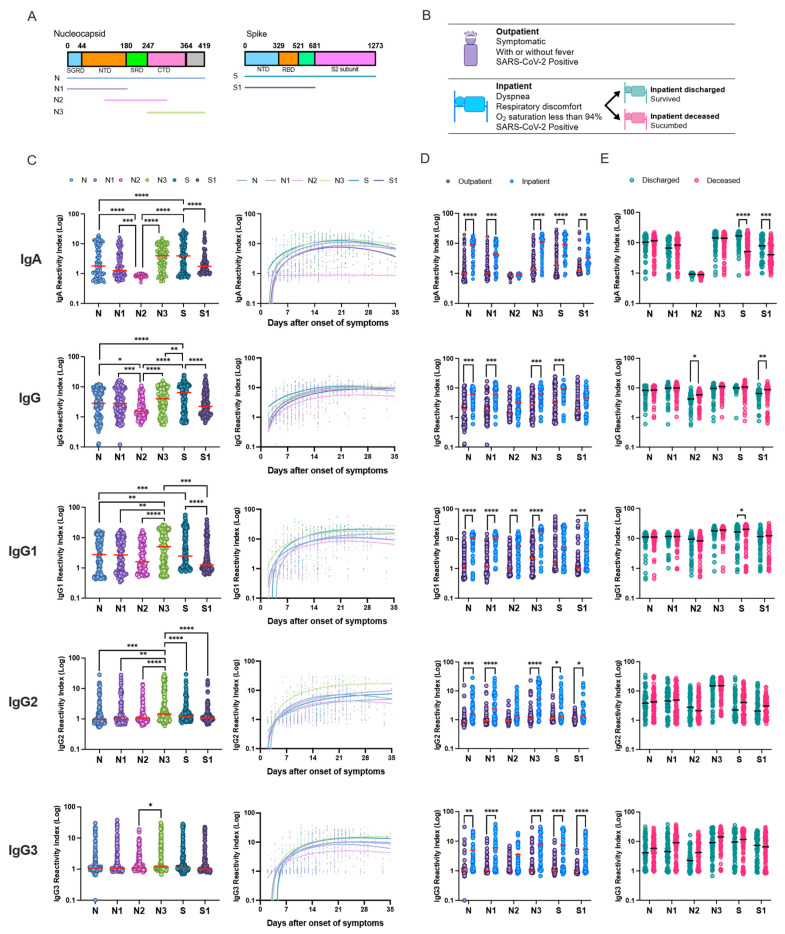
SARS-CoV-2 nucleocapsid and spike proteins elicit a robust IgA, total IgG and IgG-subclass response in COVID-19 patients. (**A**) Sketch depicts SARS-CoV-2 full-length and truncated nucleocapsid (N) and spike (S) proteins. (**B**) RT-PCR confirmed COVID-19 patients (n = 128) with non-severe disease (outpatient n = 80) and hospitalized patients with severe disease (inpatient n = 48) included in the study. (**C**–**E**) 100 ng of purified SARS-CoV-2 proteins were coated on ELISA plates and 1:100 patient serum dilution was used in an indirect ELISA to estimate serum IgA, IgG and IgG-subclass response. Antigen-specific antibody response among (**C**) all COVID-19 patients, (**D**) outpatients and inpatients, and (**E**) discharged (n = 24, samples n = 59) and deceased (n = 24, samples n = 72) inpatients was compared. Reactivity index (RI) values were calculated as a ratio between sample optical density and pre-pandemic negative samples. Horizontal red or black lines denote the median antibody range. Each curve in the graph represents reactivity towards each antigen, Splines were plotted using the Fit Spline program in GraphPad Prism software. ANOVA test with Tukey’s post hoc test was used to compare differences in the RI between antigens. Two-way ANOVA with Šidák post hoc correction for multiple comparisons. * *p* < 0.05, ** *p* < 0.01, *** *p* < 0.001, **** *p* < 0.0001.

**Figure 2 viruses-15-01018-f002:**
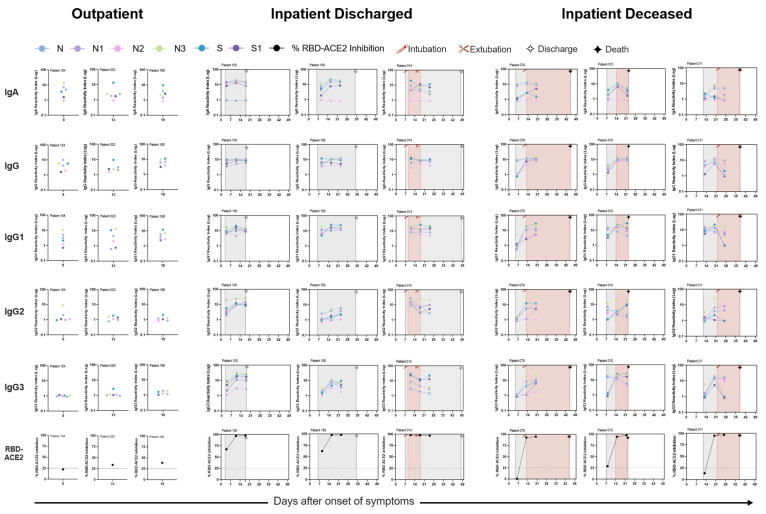
Representative COVID-19 patients and their humoral response: 100 ng of purified SARS-CoV-2 protein and 1:100 patient serum dilution was used in an indirect ELISA to estimate serum IgA, total IgG, IgG-subclass response and RBD-ACE inhibitory antibodies. Each column represents one patient, and each row represents specific response to one antibody isotype or RBD-ACE2 inhibitory antibodies. Each graph has reactivity index for SARS-CoV-2 full-length or truncated antigens tested. Outpatient samples were collected only at recruitment, whereas inpatients were followed longitudinally after hospitalization. Vertical dotted blank and red lines indicate hospitalized and intubation period, respectively (samples n = 3).

**Figure 3 viruses-15-01018-f003:**
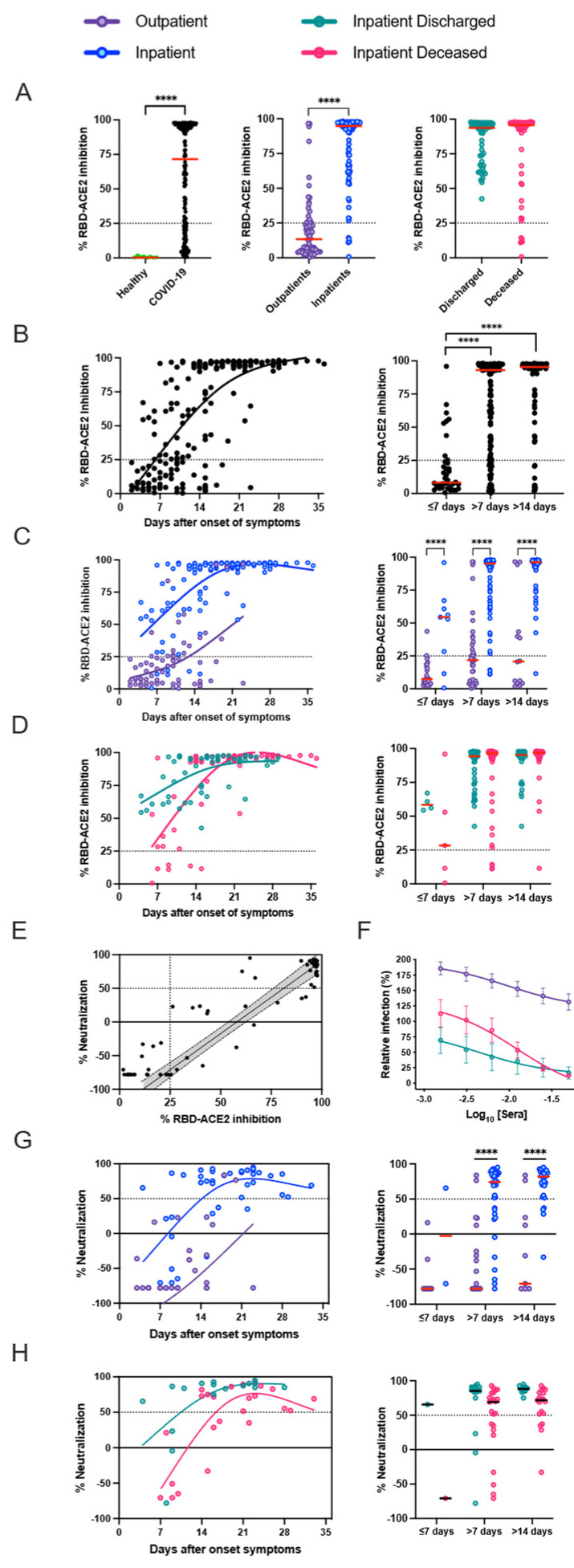
RBD-ACE2 interactions blocking antibodies and neutralizing antibody increase with symptomatic days and correlates with disease severity. (**A**) Commercial RBD-ACE2 competitive assay was used to assess inhibitory antibodies among COVID-19 patients (n = 111, samples n = 190) versus healthy individuals (n = 13). (**B**–**D**) Percentage RBD-ACE2 inhibition was evaluated post-onset of symptoms (outpatients n = 70; discharged inpatients n = 20, samples n = 54; deceased inpatients n = 21, samples n = 66). (**E**–**H**) Plasma samples were tested by PRNT in Vero cells after incubation with 300 plaque forming units (PFU) (outpatients n = 40; discharged inpatients n = 6, samples n = 18; deceased inpatients n = 8, samples n = 24). (**E**) Correlation between RBD-ACE2 competitive assay and live-virus PRNT assay (samples n = 82). (**F**) Outpatient and inpatient patient samples were serially diluted for the PRNT assay. Each data point represents the mean of all plasma samples for each group at each dilution level and error bars represent SD. (**G**,**H**) Percentage neutralization values were compared to days after onset of symptoms to evaluate neutralizing antibody kinetics. Solid lines representing the tendency for the neutralizing antibodies were created by the Fit Spline program in GraphPad Prism software. Two-way ANOVA with Šidák post hoc correction for multiple comparisons was applied. **** *p* < 0.0001.

**Figure 4 viruses-15-01018-f004:**
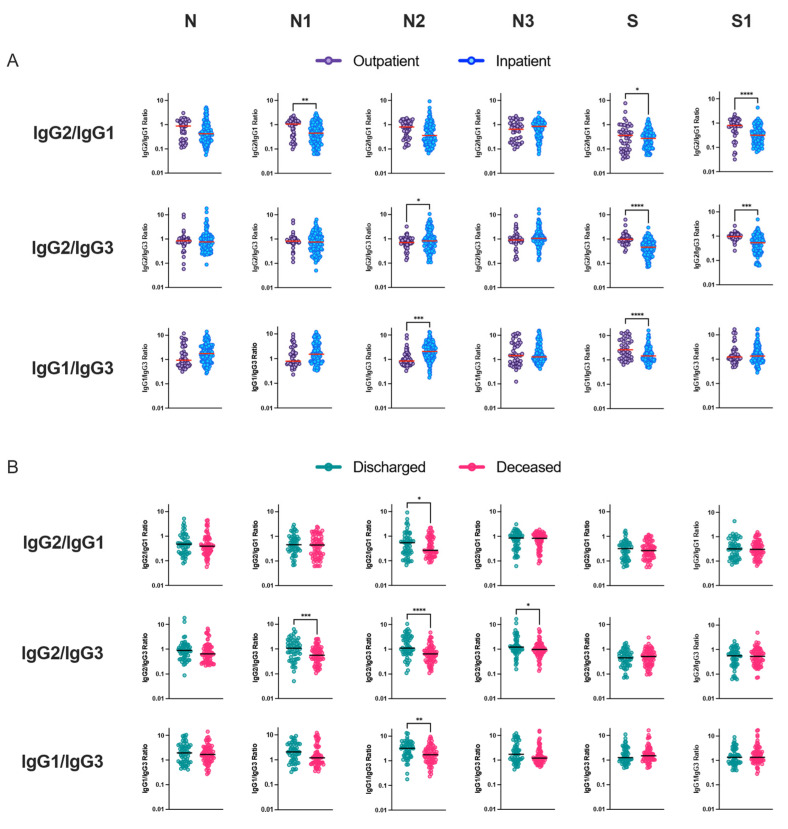
Elevated antigen-specific IgG3 response correlated with severe disease outcome. IgG subclass antibody ratio correlates with disease outcome. Ratio of full-length and truncated N and S protein-specific IgG subclass antibody response was compared between (**A**) outpatients (n = 45) and inpatients (n = 38, samples n = 121) or (**B**) hospitalized patients with discharged (n = 20, samples n = 55) or deceased (n = 18, samples n = 66) disease outcome. Samples with more than seven days after the start of symptoms were included in this analysis. T-test was performed to compare patient groups. * *p* < 0.05, ** *p* < 0.01, *** *p* < 0.001, **** *p* < 0.0001.

## Data Availability

Data is contained within the article or Appendix A.

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
