# Peer review of "Antigen-Specific Antibody Signature Is Associated with COVID-19 Outcome"

_viruses, 2023, doi:10.3390/v15041018_

Round 1

Author Response

We thank Reviewer #1 for these important comments, please check answers to the questions below.

Reviewer 2 Report

In the manuscript entitled ‘Antigen-Specific Antibody Signature Is Associated with COVID-19 Disease Outcome’ by Salgado et al., the authors investigated antibody responses against nucleocapsid and spike proteins in COVID-19 patients to find serological markers that can predict severe and non-severe patients. The authors examined IgA and IgG subclasses specific to full-length, and subdomains of N and S proteins. They found IgG1, IgG2, and IgG3 responses to N or S were significantly elevated in patients with severe diseases compared to those with non-severe diseases. They found RBD-ACE2 blocking antibodies and SARS-CoV-2 neutralizing antibodies to correlate with disease severity. Importantly, the authors found a significant difference in the IgG subclass ratio binding to N2 between discharged and deceased patients. This study was generally well-designed, and substantial serological data was generated using COVID-19 patient samples. The study is informative by finding antibody signatures for predicting COVID-19 disease severity. However, several concerns are also raised here.

Major concerns:

1.     Figures 1, 3, and 4 have no * labeled to show statistical significance. Please add them for all comparisons between groups.  

2.     Supplemental figures have low resolution and are unclear to read. Please update with a higher resolution.

3.     The authors concluded IgG subclass response towards disorganized nucleocapsid region can be used to monitor disease outcomes. Can you provide additional correlation analysis to support the sensitivity of this antibody signature?

Minor concerns:

1.     Between April to August 2020, have the SARS-CoV-2 dominating variants changed in the city of Manaus? How would the mutations affect the interpretation of the serological data in this study?    

2.     A majority of people worldwide has got COVID-19, could the antibody signature reported here be useful for predicting disease severity in future reinfection? Please discuss.

3.     Citation information for references 16 and 18 is missing.

Author Response

We thank Reviewer #2 for these important comments, please check answers to the questions below.

Reviewer 3 Report

No suggestions.

Author Response

We thank Reviewer #3 for your valuable time.